# Cardiac Structure and Function in Adults with Down Syndrome

**DOI:** 10.3390/ijerph191912310

**Published:** 2022-09-28

**Authors:** Fadi M. Azar, Victor D. Y. Beck, Alice M. Matthews, Daniel E. Forsha, Thessa I. M. Hilgenkamp

**Affiliations:** 1Kirk Kerkorian School of Medicine at University of Nevada, Las Vegas (UNLV), Las Vegas, NV 89154, USA; 2Department of Physical Therapy, University of Nevada, Las Vegas (UNLV), Las Vegas, NV 89154, USA; 3Division of Cardiology, Ward Family Heart Center, Children’s Mercy Hospital, Kansas City, MO 64108, USA

**Keywords:** Down syndrome, cardiac structure, cardiac function, echocardiography

## Abstract

Various factors may alter the risk for cardiovascular disease in adults with Down syndrome (Ds), yet few studies have examined differences in cardiac physiology in this population. Previous research suggested lower systolic and diastolic function, but inconsistent methodologies and younger samples warrant research in adults with Ds. Our aim is to compare the cardiac structure and function of adults with Ds to age- and sex-matched adults without Ds. Echocardiography was used to assess systolic function, diastolic function, and cardiac structure in *n* = 19 adults (Ds *n* = 9, control *n* = 10). Regarding cardiac structure, adults with Ds had increased left ventricular posterior wall thickness at end-systole compared to adults without Ds (*p* = 0.007). Regarding systolic and diastolic function, adults with Ds were found to have lower septal peak systolic annular velocity (S’) (*p* = 0.026), lower lateral and septal mitral annular early diastolic velocity (E’) (*p* = 0.007 and *p* = 0.025, respectively), lower lateral peak mitral annular late diastolic velocity (A’) (*p* = 0.027), and higher lateral and septal mitral annular early systolic velocity to diastolic velocity ratios (E/e’) (*p* = 0.001 and *p* = 0.001, respectively). Differences in both cardiac structure and function were found when comparing adults with Ds to matched adults without Ds. Most of the differences were indicative of worse diastolic function.

## 1. Introduction

Estimates indicate the prevalence of Down syndrome (Ds) in the United States to be 6.7 per 10,000 individuals as of 2010 [1]. Children with Ds are born with a partial or complete trisomy of chromosome 21 [2]. Congenital heart defects are present in 54% of newborn infants with Ds [3], contributing to neonatal and infant mortality rates that are 5 and 8 times greater in patients with Ds compared to the general population, respectively [4]. Significant improvements in the early recognition and repair of congenital heart defects [5,6] have led to an increase in life expectancy from 9 years in 1929 to about 60 years in the modern day [7,8,9].

Additionally, cardiac health of adults with Ds is impacted by syndrome-specific comorbidities [10]. Individuals with Ds are more likely to suffer from thyroid disease than the general population [11]. It has been suggested that both clinically apparent thyroid disease and subclinical hypothyroidism impact the cardiovascular system, leading to ventricular dysfunction and heart failure [12,13,14]. Conversely, individuals with Ds are less affected by certain pathologies that are seen more prevalently in the general population. For example, despite major risk factors for early atherosclerosis such as metabolic dysfunction [15] and obesity [16,17], adults with Ds seem to be protected against atherosclerosis [18,19,20,21]. Individuals with Ds have also been shown to exhibit lower systolic and diastolic blood pressures than individuals with other intellectual disabilities [22,23,24].

Physical inactivity may also negatively impact the cardiac health of individuals with Ds. Epidemiological evidence has indicated that sedentary behavior predicts deleterious health outcomes such as premature cardiovascular disease mortality in the general population [25,26,27,28]. One study showed an association between sedentary behavior and increased left ventricle (LV) mass in Caucasian adults [29]. Obesity is likewise positively associated with an increase in LV mass [30]. Individuals with Ds are less physically active compared to healthy controls [31,32,33,34], and 75% of females with Ds and 68% of males with Ds were either overweight or obese [35].

Current research on LV measurements and characteristics in the Ds population are both scarce and inconsistent, highlighting a need for further investigation. A study by Vis et al. showed that individuals with Ds had significantly lower LV mass and LV volume compared to healthy controls. LV diameters were not significantly different in the child population of this study, indicating that the differences in the adult population were acquired and could be related to inactivity [36]. A study conducted by Kelly et al. described contrary results, reporting that youth and young adults with Ds had decreased LV size, as well as decreased diastolic functions, indicating stiffer LV. No significant difference in LV ejection fraction was appreciated between the two groups [9]. A previous study conducted by Balli et al. reported further conflicting results, showing that children with Ds had both increased LV mass and LV ejection fraction compared to children without Ds, along with diastolic dysfunction measured through Doppler imaging [37]. Both the scarcity of current research on this subject and inconsistencies in the available results make further investigation imperative. The small sample size in the Balli et al. study [37] and relatively young cohort in the Kelly et al. study [9] indicate a need for further exploration of properties of both cardiac muscle morphology and function of individuals with Ds, particularly in the adult population. Therefore, the primary purpose of this study is to compare the structural and functional properties of the hearts of adults with Ds with those of control participants without Ds. We hypothesize that differences in cardiac structure and function within the adult Ds population are indicative of low levels of physical activity, high levels of obesity, and underlying cardiac pathology within the Ds population.

## 2. Materials and Methods

### 2.1. Design and Participants

This cross-sectional analysis compared echocardiogram images of individuals with Ds between the ages of 18 and 45 years old with those of individuals without Ds matched for age, sex, race, and body mass index. This study is part of larger study investigating blood flow regulation in individuals with Ds.

All of the study procedures were conducted in the Integrated Physiology Laboratory at the University of Illinois at Chicago. Individuals with Ds and individuals without Ds were recruited in the Chicago, IL area through the use of flyers, advertisements in local newspapers, electronic communication, social media, newsletters, word of mouth, and engagement with support groups and organizations that serve individuals with Ds. To be enrolled into the study, individuals or their caregivers were required to initiate contact with the Integrative Physiology Laboratory in order to obtain further information about the study and to be screened for eligibility. 

Inclusion criteria for this study required that individuals were between the ages of 18 and 45 years old, generally healthy, yet inactive. ‘Inactive’ was described as being involved in less than 30 min of moderately intense physical activity per day on at least 5 days/week. Physical activity was assessed through use of the International Physical Activity Questionnaire (IPAQ) Short Form [38]. Participants with Ds were required to have a diagnosis of Ds trisomy 21, as well as normal or stable thyroid function for at least 6 months. 

Exclusion criteria for this study were any type of current known cardiac disease including valvular disease, atherosclerotic or other vascular diseases, asthma or other pulmonary diseases, hypertension (defined as a blood pressure greater than 140/90 mmHg), and a history of pre-syncope or syncope. Further exclusion criteria included diabetes (defined as an HbA1c of greater than 7.5% or use of glucose lowering medication), severe obesity (defined as a body mass index of greater than 40), current smoking, and pregnancy. Individuals taking anti-inflammatory medications (including nonsteroidal anti-inflammatory drugs) or medication affecting heart rate, blood pressure, or arterial function were also excluded from this study.

### 2.2. Procedure

Participants and their care givers provided written informed consent. Participants, as well as their parents and care givers, were familiarized with the laboratory equipment and procedures prior to the commencement of data collection. Women of childbearing age were tested for pregnancy through the use of a urine pregnancy test. Female participants were studied during the first 3–5 days of menses. Females taking oral contraceptive medication were studied during the placebo phase.

Participants were tested in a postprandial state, at least three hours after their last meal. Participants were asked to refrain from any exercise, caffeine consumption, or alcohol consumption during the 24-hour period preceding their study visit. The height, weight, and circumference of the participants were measured using standard techniques. Blood pressure measurements were obtained through an automated sphygmomanometer, including systolic blood pressure (SBP), diastolic blood pressure (DBP), and mean arterial pressure (MAP). Resting heart rate (HR) was also obtained using standard technique.

Two-dimensional echocardiography focused on the structure and function of the left ventricle was conducted on participants using a Hitachi Aloka Alpha 7 system (Tokyo, Japan). Participants were asked to lay in the supine and left lateral decubitus position. Measurements were obtained using the parasternal long-axis, parasternal short-axis, and four-chamber apical view. The B-mode and M-mode functions of the ultrasound device were utilized to obtain the measurements. Heart rate was measured through use of a standard electrocardiogram with 3-lead electrocardiogram. The electrodes were interfaced with the ultrasound machine to collect the data. Three consecutive cardiac cycles were analyzed.

### 2.3. Outcome Measures

LV size was obtained through measurement of LV internal cavity diameters at both end-diastole and end-systole using two-dimensional echocardiographic imaging [39]. Both the parasternal long axis and parasternal short axis views were obtained. The following LV internal cavity diameters were measured through manual tracing: the intraventricular septal diastolic thickness (IVSd), left ventricle internal diameter at end-diastole (LVIDd), left ventricle posterior wall diastolic thickness (LVPWd), intraventricular septal systolic thickness (IVSs), left ventricle internal diameter at end-systole (LVIDs), and left ventricle posterior wall systolic thickness (LVPWs). 

Simpson’s biplane method [40,41] was used to calculate the following volumes from the four chamber apical view (LV4): end diastolic volume (EDV), end systolic volume (ESV), ejection fraction (LVEF), fractional shortening (FS), and left ventricular mass (LVM).

Mitral inflow (MF) was observed in the apical view to assess the ratio of early to late diastolic transmitral flow velocity (E/A). Tissue Doppler imaging (TDI) in the LV4 view was used to measure lateral and septal peak systolic annular velocity (S’), lateral and septal mitral annular early diastolic velocity (E’), lateral and septal peak mitral annular late diastolic velocity (A’), ratios of lateral and septal mitral annular early diastolic velocity to mitral annular late diastolic velocity (E’/A’), and lateral and septal mitral annular early systolic velocity to diastolic velocity ratios (E/e’).

The following parameters correspond to cardiac systolic function: LVEF, FS, and S’ [42]. The following parameters correspond to cardiac diastolic function: E/A, E’, A’, E’/A’, and E/e’ [43]. The following parameters correspond to cardiac structure: IVSs, IVSd, LVPWs, LVPWd, LVIDs, LVIDd, EDV, ESV, and LVM [44].

### 2.4. Statistical Analysis

Data were checked for normal distribution through calculations of skewness and kurtosis and Shapiro–Wilk test. Data are reported as means and standard deviations (normally distributed variables) or medians and interquartile ranges (non-normally distributed variables). Differences in continuous baseline variables between participants with Ds and matched controls were checked through independent *t*-tests if normally distributed, or with Mann–Whitney U tests if not normally distributed.

Statistical analyses were carried out using the IBM SPSS Version 27 statistical software (Chicago, IL, USA) for Macintosh. The level of significance was set at *p* < 0.05 for all analyses.

## 3. Results

Nine participants with Ds and ten participants without Ds were included in this study. The characteristics of the adults with Ds and the adults without Ds are shown in Table 1. Statistically significant differences were observed in both height and BMI between the two groups (*p* < 0.01). No statistically significant differences were seen in age or weight (*p* > 0.05).

Findings based on echocardiography are displayed in Table 2. LVPWs was significantly thicker in adults with Ds (*p* < 0.01). S’, lateral and septal E’, and lateral A’ were all higher in the adults without Ds (*p* < 0.05). Both the lateral and septal E/e’ were higher in adults with Ds compared to adults without Ds (*p* < 0.01).

## 4. Discussion

In this study, we utilized echocardiography to measure the structure and function of the heart in adults with Ds and without Ds. We identified statistically significant differences in measurements that indicate a potential reduction in diastolic function in adults with Ds compared to adults without Ds, demonstrated by significantly lower values in E’ and A’, along with significantly higher values of E/e’ in the participants with Ds. 

### 4.1. Diastolic Function in Adults with Ds

Our study identified increased E/e’ ratios in participants with Ds compared to participants without Ds. While invasive measures of diastolic function obtained in the cardiac catheterization lab are the gold standard, multiple studies have demonstrated that the echocardiographic measure of E/e’ ratio correlates with left ventricular filling pressure, a non-invasive measure of diastolic function [45,46]. The between-group difference in E/e’ indicates a potential abnormality in diastolic function in adults with Ds. The study by Kelly et al. which concentrated on adolescents and young adults with Ds found a statistically significant increase in E/e’ in the Ds group [9]. This finding is consistent with our finding of increases in both lateral and septal E/e’ in adults with Ds. While this difference in diastolic function could be genetically driven, the increased BMI in the group with Ds or other unmeasured factors such as possible differences in physical activity levels could also play a role.

Another non-invasive marker of diastolic dysfunction is E/A ratio [45]. A decrease in diastolic function was observed in the study conducted by Balli et al., which found a statistically significant difference in early to late transmitral flow velocity (E/A) ratio between children with Ds and children without Ds [37]. Our study did not observe a statistically significant difference in E/A ratio between the two groups. However, E/A ratio has a complex and non-linear relationship with the severity of diastolic dysfunction complicating the interpretation. In the middle stages of progression of diastolic disease, the E/A ratio normalizes and could explain variable results in different studies for this measure [45]. 

### 4.2. Systolic Function in Adults with Ds

Overall, systolic function was similar between the two groups in this study as measured by LV ejection fraction and shortening fraction, two traditional measures of global LV systolic function [41]. Our results did show that both these measures trended slightly lower in the group with Ds, but this difference was not statistically significant and they remained in the normal range in both groups. The adults with Ds did show lower mitral annular septal systolic velocity (S’) on the septal aspect, suggestive of slightly diminished contractility of the septal myocardium. This is an interesting finding that, to our knowledge, has not been reported in the Ds population previously. Should these subtle changes in function or hypertrophy in the Ds progress, they can lead to heart failure, poor quality of life, and early morbidity and mortality. Further research is necessary to determine if there may be genetic factors involved that would explain this finding, and longitudinal studies are required to assess whether these changes are progressive over time.

### 4.3. Cardiac Structure in Adults with Ds

The only structural measure that was significantly different between groups was LVPWs. Evidence of cardiac hypertrophy was present for the Ds group in the LV posterior wall thickness measured at end-systole but was not significant for the end-diastolic measure, likely representing an early/subtle phase of LV hypertrophy. Otherwise, measures of cardiac wall thickness and size were not significantly different in the group with Ds. Left ventricular hypertrophy or thickening of the LV wall may result from chronic hypertension in other populations [47] but there was no difference in the blood pressure between the groups. Furthermore, adults with Ds tend to have lower blood pressure on average than the typically developed population [24]. We can only speculate about other causes, such as a possible obesity-related (not hypertension-mediated), or a Ds genetic related change in the structure of the myocardium, but the scarcity of research on this topic prevents clear guidance, and further research is greatly needed. A study conducted by Vis et al. found that adults with intellectual disabilities, including a large group of Ds, have significantly smaller cardiac size and LV dimensions. The results of that aforementioned study were not consistent with our study population [36]. While their study population included those with Ds, it was a heterogenous group that is not ideally comparable to this current study population. Further investigation is needed to further explore differences in cardiac structure in adults with Ds.

### 4.4. Limitations

A significant limitation of this exploratory study is the relatively small sample size of the two study groups due to difficulties in recruitment, reducing its power substantially. The COVID-19 pandemic significantly hindered our ability to recruit additional participants for the study due to university research policies and general hesitancy to participate during the pandemic for this high-risk population [48]. A second limitation is that we do not have data on the right ventricle, and certain parameters may be influenced by the right ventricle. The ventricular-ventricular interactions in adults with Ds would be an important step for future research.

### 4.5. Future Considerations

This study suggests that a reduction in diastolic function may be seen in adults with Ds, a finding that has been detailed by previous investigations on this population [9,37]. Nonetheless, due to the paucity of literature that explores this topic, further studies analyzing cardiovascular differences in adults with Ds in larger sample sizes and that include a detailed description of their history of congenital heart defects and surgeries are required. Longitudinal studies are needed to assess progression over time of the subtle changes towards more serious health concerns such as heart failure and early morbidity and mortality. Further exploration regarding interventions that may improve cardiac function in adults with Ds is also required. Various studies have shown that exercise training can significantly improve diastolic function and quality of life in individuals with heart failure [49,50]. It is important to explore the use of exercise and other medical interventions to further improve the lifespan of adults with Ds and their overall quality of life. Additionally, it is important for future studies to evaluate the association between cardiac function in adults with Ds, obesity, and the sedentary lifestyle of many individuals in that population. The interplay of having Ds, obesity/overweight and various activity levels on diastolic function needs to be unraveled in studies including both inactive and active adults with Ds of different weight status. Through the identification of differences in the cardiac structure and function of adults with Ds compared with adults without Ds, both researchers and clinicians may begin to better understand the factors that underlie cardiovascular morbidity and mortality in this population.

## 5. Conclusions

In summary, this study has identified differences in cardiac parameters in adults with Ds suggesting altered global LV diastolic function and regional abnormalities in systolic function and structure. These findings contribute to our understanding of cardiac structure and function in adults with Ds, a major contributor to their overall health and wellbeing. Through further evaluation of cardiac health in adults with Ds, progress can be made toward further reducing cardiac morbidity and mortality in this population of individuals.

## Figures and Tables

**Table 1 ijerph-19-12310-t001:** Participant Characteristics.

	Down Syndrome (n = 9)	Control (*n* = 10)	*p*-Value
Age (y) ^b^	25.5 (5.50)	23.5 (5.25)	0.393
Height (cm) ^a^	160.55 ± 7.96	171.65 ± 7.08	0.004 *
Weight (kg) ^a^	77.14 ± 13.67	69.39 ± 14.44	0.234
BMI (kg/m^2^) ^a^	29.95 ± 4.74	23.46 ± 3.99	0.004 *
MAP (mmHg) ^a^	85.24 ± 8.4	86.69 ± 7.6	0.708
SBP (mmHg) ^a^	120.28 ± 6.1	117.28 ± 7.9	0.379
DBP (mmHg) ^a^	67.72 ± 11.5	71.39 ± 8.4	0.451
RHR (bpm) ^a^	60.0 ± 10.1	65.3 ± 7.0	0.189

Note: Values are presented as mean ± SD or median (IQR); ^a^: differences tested with an independent *t*-test; ^b^: differences tested with a Mann–Whitney U test; * Significantly different; *p* < 0.05; BMI—body mass index; MAP—mean arterial pressure; SBP—systolic blood pressure; DBP—diastolic blood pressure; RHR—resting heart rate.

**Table 2 ijerph-19-12310-t002:** Echocardiograph Findings.

	Down Syndrome (*n* = 9)	Control (*n* = 10)	*p*-Value
Systolic Function			
LVEF	56.00 ± 11.04	58.39 ± 11.88	0.630
FS	25.97 ± 4.87	26.29 ± 7.83	0.917
S’ TDI Lateral	0.09 ± 0.03	0.09 ± 0.01	0.657
S’ TDI Septal	0.08 ± 0.01	0.09 ± 0.01	0.026 *
Diastolic Function			
E/A Mitral Flow ^a^	2.65 (1.52)	2.24 (0.45)	0.447
E’ TDI Lateral	0.13 ± 0.02	0.15 ± 0.02	0.007 *
E’ TDI Septal	0.11 ± 0.02	0.13 ± 0.02	0.025 *
A’ TDI Lateral	0.06 ± 0.01	0.08 ± 0.02	0.027 *
A’ TDI Septal	0.07 ± 0.02	0.08 ± 0.02	0.063
E’/A’ TDI Lateral	2.30 ± 0.52	2.11 ± 0.67	0.513
E’/A’ TDI Septal ^a^	1.81 (0.59)	1.73 (0.69)	0.633
E/e’ TDI Lateral	7.24 ± 1.81	4.66 ± 0.97	0.001 *
E/e’ TDI Septal ^a^	8.24 (2.81)	4.92 (2.16)	0.001 *
Cardiac Structure			
IVSs	1.03 ± 0.15	1.02 ± 0.13	0.937
IVSd ^a^	0.80 (0.10)	0.76 (0.16)	0.720
LVPWs	1.42 ± 0.14	1.24 ± 0.12	0.007 *
LVPWd ^a^	0.84 (0.13)	0.80 (0.08)	0.133
LVIDs ^a^	3.05 (0.69)	2.77 (1.17)	0.447
LVIDd ^a^	4.29 (0.76)	4.21 (1.20)	0.604
LVM ^a^	104.00 (34.00)	90.50 (82.63)	0.447

Note: Values are presented as mean ± SD, differences tested with an independent *t*-test; ^a^: Values presented as median (IQR), differences tested with a Mann–Whitney U test; * Significantly different; *p* < 0.05; LVEF—ejection fraction; FS—fractional shortening; S’—lateral and septal peak systolic annular velocity; TDI—tissue Doppler imaging; E/A—the ratio of early to late diastolic transmitral flow velocity; E’—lateral and septal mitral annular early diastolic velocity; A’—lateral and septal peak mitral annular late diastolic velocity; E’/A’—ratio of lateral and septal mitral annular early diastolic velocity to mitral annular late diastolic velocity; E/e’—lateral and septal mitral annular early systolic velocity to diastolic velocity ratio; IVSs—intraventricular septal systolic thickness; IVSd—intraventricular septal diastolic thickness; LVPWs—left ventricle posterior wall systolic thickness; LVPWd—left ventricle posterior wall diastolic thickness; LVIDs—left ventricle internal diameter at end-systole; LVIDd—left ventricle internal diameter at end-diastole; LVM—left ventricular mass.

## Data Availability

The data presented in this study are available on request from the corresponding author. The data are not publicly available due to concerns for the privacy of all participants.

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
