# Peer review of "Cardiac Structure and Function in Adults with Down Syndrome"

_ijerph, 2022, doi:10.3390/ijerph191912310_

Round 1
Reviewer 1 Report
The policy of International Journal of Environmental Research and and Public Health includes issues regarding environmental sciences and engineering, public health, environmental health, occupational hygiene, health economic and global health research. The authors clearly described that differences in both cardiac structure and function were found when comparing adults with Down syndrome to matched control subjects. However, the described issue is unlikely to be suitable to the IJERPH. In addition, the number of subjects are too small to draw the conclusion.
Author Response
Response to Reviewer 1 Comments
Comment 1: The policy of International Journal of Environmental Research and Public Health includes issues regarding environmental sciences and engineering, public health, environmental health, occupational hygiene, health economic and global health research. The authors clearly described that differences in both cardiac structure and function were found when comparing adults with Down syndrome to matched control subjects. However, the described issue is unlikely to be suitable to the IJERPH. In addition, the number of subjects is too small to draw the conclusion.
Response 1: We thank the reviewer for their consideration of the topic for the International Journal of Environmental Research and Public Health. However, the special issue on “Diseases Etiology and Management: Towards a Precision Medicine Approach” would be a great fit for our research. We agree that our sample size is small and finding significant differences in such a small sample is even more challenging. In addition to acknowledging the limitations of this small sample size in the Limitations section, we have also emphasized the need for confirming these results in future studies with larger sample sizes in the section ‘Future Considerations’.
Reviewer 2 Report
The authors evaluated several aspects regarding the LV structure as well as the LV systolic and diastolic function in patients suffering from Down syndrome. They found significant differences mainly concerning the LV diastolic function when compared to age and sex-matched adults without Ds. The manuscript is well-written and it addresses an important health issue, offering clear and interesting results. However, I have some comments/recommendations:
1. The authors should state whether there was any significant mitral valve/other valvular disease and/or any recent cardiac intervention that might have influenced the results.
2. For the systolic function, it would have been interesting to evaluate the LV strain by using the speckle tracking technique (at least the longitudinal strain, if not radial and/or circumferential strain) as it offers a better insight into this area.
3. The lower mitral annulus S' may also reflect a right ventricular problem. I would recommend for the authors to offer some Information about the RV size and function, as well.
4. For relevance mathers, I would recommend for the authors to state their opinion about the fact that în their study only the LVPWs thickness was significantly different between groups and not LVPWd and IVS, as well.
Author Response
Response to Reviewer 2 Comments
The authors evaluated several aspects regarding the LV structure as well as the LV systolic and diastolic function in patients with Down syndrome. They found significant differences mainly concerning the LV diastolic function when compared to age and sex-matched adults without Ds. The manuscript is well-written and it addresses an important health issue, offering clear and interesting results. However, I have some comments/recommendations:
- The authors should state whether there was any significant mitral valve/other valvular disease and/or any recent cardiac intervention that might have influenced the results.
Response 1: Thank you for pointing this out. None of the participants had any current mitral valve or valvular disease nor any recent cardiac intervention. For clarity, we have added the following to the description of the exclusion criteria: “Exclusion criteria for this study were any type of current known cardiac disease including valvular disease, ….” - For the systolic function, it would have been interesting to evaluate the LV strain by using the speckle tracking technique (at least the longitudinal strain, if not radial and/or circumferential strain) as it offers a better insight into this area.
Response 2: We agree with the reviewer that this would be interesting, but as these images were not optimized for strain imaging, we felt these measurements would be suboptimal. We will keep this in mind for our future studies.
- The lower mitral annulus S' may also reflect a right ventricular problem. I would recommend for the authors to offer some Information about the RV size and function, as well.
Response 3: We agree with the reviewer that this would indeed be very interesting, however, our protocol did not include these measures. To set the correct expectations, we have now added to the methods section that our protocol focused on the left ventricle, and we have acknowledged this as a limitation in the discussion: “A second limitation is that we do not have data on the right ventricle, and certain parameters may be influenced by the right ventricle. The ventricular-ventricular interactions in adults with Ds would be an important step for future research.”.
- For relevance mathers, I would recommend for the authors to state their opinion about the fact that în their study only the LVPWs thickness was significantly different between groups and not LVPWd and IVS, as well.
Response 4: Thank you for inviting our opinion on this matter, we have added the following to the section on Cardiac Structure in the Discussion: “Evidence of cardiac hypertrophy was present for the Ds group in the LV posterior wall thickness measured at end-systole but was not significant for the end-diastolic measure, likely representing an early/subtle phase of LV hypertrophy.”
Reviewer 3 Report
The study by Azar et al. aimed to characterize cardiac structure function in individuals with Ds in comparison to healthy age and sex matched adults. This is cited as a novel research question, in that existing literature in this area is inconsistent and there needs to be more research on the characteristics of Ds adults vs. their younger counterparts. The researchers hypothesized that differences in cardiac structure and function would be attributed to low levels of physical activity, obesity and cardiac pathology in the Ds cohort. Main findings displayed via echocardiography were possible differences in diastolic function (higher E/e’ ratios), systolic function (S’), and cardiac structure (LVPWs). The manuscript is well-written and acknowledges the previous literature relevant to the findings of this manuscript. Overall, the characterization of Ds in this capacity is novel and important, but does continue to leave questions due a continued disconnect with the findings of previous literature. Below are a few concerns/suggestions that the authors might want to consider:
1. The authors hypothesized low levels of physical activity and high levels of obesity would contribute to the differences in cardiac structure/function (71-73), but the way in which these things are defined and brought up throughout the manuscript lead to a few thoughts:
a. “inactive” is described as less than 30 min of moderate intensity exercise per day (89-90). Is there a reason that this definition is used? Seemingly, one could participate in moderate intensity exercise below this time threshold each day and still actually meet the recommended PA guidelines by NIH/ACSM/etc. Could these individuals really then be considered to be inactive? Authors go on to discuss exercise training impacts on improved diastolic function and QoL (253-254). Is there a potential that some of the disparities in diastolic function (ex: E/A ratio; 209-216) from other studies be due to a difference in Ds activity level where some dysfunction was masked in this study?
b. BMI seems to be how overweight/obesity is defined in this study, and there is a significant difference in BMI between Ds and controls (table 1). However, there are no significant weight differences reported between groups and BMI might then be driven by the significant height differences instead. Additionally, there is literature to suggest BMI is not necessarily the best way to determine overweight/obese health status due to differences in sex/race/etc. Authors cite increased BMI and/or PA levels (206-208) as the potential reason for E/e’ differences, but could there be other explanations? Based off weight and previous questions about “inactivity”, it would seem like Ds was not any more overweight than controls and there could be some question about if some individuals could actually be classified as active.
2. LVPW was significantly different between groups, and the authors cite that issues could arise as a result of chronic hypertension. I believe more of a discussion around the differences in LVPW would be helpful. As the authors state (230-240), chronic hypertension is not something normally seen with Ds and in fact studies show much lower BP in Ds and a cardioprotective nature (38-44). Interestingly, this study did not show a significant difference in BP between groups as they have mentioned is often found in the literature. Can more discussion be offered regarding why LVPW might be different and how this can be reconciled with similar BPs reported?
3. Further characterization of the cardiac structure/function is very important, and I commend the work this group has done to add the literature. I also appreciate the authors reporting clearly how their work aligns with what has previously been reported. Could the authors also add a few statements that also describe how increases/decreases of their significant measures would directly impact Ds health? A comment in the conclusion alludes to the importance of understanding the differences because it could be a major contributor to overall health and wellbeing (266-168), so direct/specific examples of that throughout the manuscript would be helpful.
Minor:
1. Can you clarify exclusion of unresolved congenital heart disease (95), did some of your participants still have defects that were left untreated because they were not serious? Or was there no presence at all? This would also help clarify statements made in future considerations (251-252) that states considering congenital heart defects in adults is required in the future.
2. Reference #11 in the list needs to be fixed.
Author Response
Response to Reviewer 3 Comments
The study by Azar et al. aimed to characterize cardiac structure function in individuals with Ds in comparison to healthy age and sex matched adults. This is cited as a novel research question, in that existing literature in this area is inconsistent and there needs to be more research on the characteristics of Ds adults vs. their younger counterparts. The researchers hypothesized that differences in cardiac structure and function would be attributed to low levels of physical activity, obesity and cardiac pathology in the Ds cohort. Main findings displayed via echocardiography were possible differences in diastolic function (higher E/e’ ratios), systolic function (S’), and cardiac structure (LVPWs). The manuscript is well-written and acknowledges the previous literature relevant to the findings of this manuscript. Overall, the characterization of Ds in this capacity is novel and important, but does continue to leave questions due a continued disconnect with the findings of previous literature. Below are a few concerns/suggestions that the authors might want to consider:
- The authors hypothesized low levels of physical activity and high levels of obesity would contribute to the differences in cardiac structure/function (71-73), but the way in which these things are defined and brought up throughout the manuscript lead to a few thoughts:
1a. “inactive” is described as less than 30 min of moderate intensity exercise per day (89-90). Is there a reason that this definition is used? Seemingly, one could participate in moderate intensity exercise below this time threshold each day and still actually meet the recommended PA guidelines by NIH/ACSM/etc. Could these individuals really then be considered to be inactive? Authors go on to discuss exercise training impacts on improved diastolic function and QoL (253-254). Is there a potential that some of the disparities in diastolic function (ex: E/A ratio; 209-216) from other studies be due to a difference in Ds activity level where some dysfunction was masked in this study?
Response to 1a: We apologize for any confusion by leaving out important information, we have added ‘on at least 5 days per week’ to the description of ‘less than 30 minutes of moderate intensity exercise per day’, which was how our eligibility criterium was formulated, and which was deliberately chosen to align with the physical activity guidelines from the U.S. Department of Health and Human Services and other guidelines. By keeping physical activity levels similar between the groups, we avoided differences in cardiac structure and function attributable to different physical activity levels. We appreciate the reviewers’ remark on the influence of inactivity on diastolic function, and future studies including both inactive and active individuals with Down syndrome are needed to unravel these interactions. Also incorporating our response to the comment 1b, we have added this to the section on Future Considerations: “The interplay of having Ds, obesity/overweight and various activity levels on diastolic function needs to be unraveled in studies including both inactive and active adults with Ds of different weight status.”
1b. BMI seems to be how overweight/obesity is defined in this study, and there is a significant difference in BMI between Ds and controls (table 1). However, there are no significant weight differences reported between groups and BMI might then be driven by the significant height differences instead. Additionally, there is literature to suggest BMI is not necessarily the best way to determine overweight/obese health status due to differences in sex/race/etc. Authors cite increased BMI and/or PA levels (206-208) as the potential reason for E/e’ differences, but could there be other explanations? Based off weight and previous questions about “inactivity”, it would seem like Ds was not any more overweight than controls and there could be some question about if some individuals could actually be classified as active.
Response to 1b: We agree with the reviewer that more precise measures like a DEXA scan would be preferred to determine overweight/obese health status, but BMI was the only data collected in this study protocol on obesity status. Although weight is not significantly different between groups, due to the smaller statue the individuals with Ds are indeed more overweight, as reflected by BMI, and consistent with other measures in other studies in individuals with Ds. However, the goal of our study was to compare cardiac function between individuals without Ds and with Ds, and not to determine the impact of either physical activity levels or BMI on cardiac function, as we agree with the reviewer that this would require more detailed measures of each of those determinants and a larger sample size to accommodate those analyses. We included the group characteristics as context only, and would prefer to avoid speculation on determinants of our findings beyond the description in our current discussion. Instead, we have added the study of determinants to our Future Considerations “The interplay of having Ds, obesity/overweight and various activity levels on diastolic function needs to be unraveled in studies including both inactive and active adults with Ds of different weight status.”
- LVPW was significantly different between groups, and the authors cite that issues could arise as a result of chronic hypertension. I believe more of a discussion around the differences in LVPW would be helpful. As the authors state (230-240), chronic hypertension is not something normally seen with Ds and in fact studies show much lower BP in Ds and a cardioprotective nature (38-44). Interestingly, this study did not show a significant difference in BP between groups as they have mentioned is often found in the literature. Can more discussion be offered regarding why LVPW might be different and how this can be reconciled with similar BPs reported?
Response 2. We agree with the reviewer that this is indeed very interesting, and we have cautiously included some thoughts in the Discussion: “We can only speculate about other causes, such as a possible obesity-related (not hypertension-mediated), or a Ds genetic related change in the structure of the myocardium, but the scarcity of research on this topic prevents clear guidance, and further research is greatly needed.”
- Further characterization of the cardiac structure/function is very important, and I commend the work this group has done to add the literature. I also appreciate the authors reporting clearly how their work aligns with what has previously been reported. Could the authors also add a few statements that also describe how increases/decreases of their significant measures would directly impact Ds health? A comment in the conclusion alludes to the importance of understanding the differences because it could be a major contributor to overall health and wellbeing (266-168), so direct/specific examples of that throughout the manuscript would be helpful.
Response 3: Thank you for pointing this out, we have now included information about what progression of these subtle changes could mean in the section on Systolic function: “Should these subtle changes in function or hypertrophy in the Ds progress, they can lead to heart failure, poor quality of life, and early morbidity and mortality.”, and in the section on future considerations we have included a recommendation for longitudinal studies incorporating this information: “Longitudinal studies are needed to assess progression over time of the subtle changes towards more serious health concerns such as heart failure and early morbidity and mortality.”
Minor:
- Can you clarify exclusion of unresolved congenital heart disease (95), did some of your participants still have defects that were left untreated because they were not serious? Or was there no presence at all? This would also help clarify statements made in future considerations (251-252) that states considering congenital heart defects in adults is required in the future.
Response minor comment 1: None of the participants had current defects that were left untreated, but we did not collect detailed information on their history of congenital heart disease and/or surgeries. We have clarified our exclusion criteria: “Exclusion criteria for this study were any type of current known cardiac disease including valvular disease, ….”, and to avoid any confusion, we have also clarified the section in the Discussion: ““further studies analyzing cardiovascular differences in adults with Ds in larger sample sizes and that include a detailed description of their history of congenital heart defects and surgeries are required.”
- Reference #11 in the list needs to be fixed.
Response minor comment 2: Thank you for pointing this out, we have corrected this.
Round 2
Reviewer 1 Report
This manuscript has been successfully revised. Thank you.